# DNA Sequence Variations Affecting Serotonin Transporter Transcriptional Regulation and Activity: Do They Impact Alcohol Addiction?

**DOI:** 10.3390/ijms25158089

**Published:** 2024-07-25

**Authors:** Giampiero Ferraguti, Silvia Francati, Claudia Codazzo, Giovanna Blaconà, Giancarlo Testino, Antonio Angeloni, Marco Fiore, Mauro Ceccanti, Marco Lucarelli

**Affiliations:** 1Department of Experimental Medicine, Sapienza University of Rome, 00161 Rome, Italy; silvia.francati@uniroma1.it (S.F.); giovanna.blacona@uniroma1.it (G.B.); giancarlo.testino@uniroma1.it (G.T.); antonio.angeloni@uniroma1.it (A.A.); marco.lucarelli@uniroma1.it (M.L.); 2UOSD Genetica Medica, Asl Roma1, 00193 Roma, Italy; claudia.codazzo@gmail.com; 3Institute of Biochemistry and Cell Biology, IBBC-CNR, 00161 Rome, Italy; marco.fiore@cnr.it; 4SITAC, Società Italiana Per il Trattamento Dell’alcolismo e le Sue Complicanze, 00185 Rome, Italy; mauro.ceccanti@uniroma1.it; 5Pasteur Institute, Cenci Bolognetti Foundation, Sapienza University of Rome, 00161 Rome, Italy

**Keywords:** alcohol dependence, serotonin transporter, polymorphisms, molecular biology, gene expression

## Abstract

Genetic features of alcohol dependence have been extensively investigated in recent years. A large body of studies has underlined the important role of genetic variants not only in metabolic pathways but also in the neurobiology of alcohol dependence, mediated by the neuronal circuits regulating reward and craving. Serotonin transporter (5-HTT), encoded by the SLC6A4 gene (Solute carrier family 6-neurotransmitter transporter-member 4), is targeted by antidepressant drugs such as selective serotonin reuptake inhibitors (SSRIs) and plays a pivotal role in serotoninergic transmission; it has been associated with psychiatric diseases and alcohol dependence. Transcriptional regulation and expression of 5-HTT depend not only on epigenetic modifications, among which DNA methylation (CpG and non-CpG) is primarily involved, but also on sequence variations occurring in intron/exon regions and in untranslated regions in 5′ and 3′, being the first sequences important for the splicing machinery and the last for the binding of transcription factors and micro RNAs. This work intends to shed light on the role of sequence variations known to affect the expression or function of 5-HTT in alcohol-dependent individuals. We found a statistically significant difference in the allelic (*p* = 0.0083) and genotypic (*p* = 0.0151) frequencies of the tri-allelic polymorphism, with higher function alleles and genotypes more represented in the control population. Furthermore, we identified three haplotypes more frequent in subjects with AUD (*p* < 0.0001) and one more frequent in the control population (*p* < 0.0001). The results obtained for the tri-allelic polymorphism in alcohol dependence confirm what is already present in part of the literature. The role of haplotypes requires further studies to be clarified.

## 1. Introduction

Alcohol consumption is the primary emerging risk factor when compared to all the other illegal substances of abuse, in terms of perpetrating violent acts, crimes, familial harassment, memory and learning impairments, loss of productivity and, last but not least, augmented susceptibility to infectious diseases [1,2,3,4,5,6,7,8,9,10]. Italy has one of the lowest prevalences of alcohol use disorders (AUDs) in Europe for both men (1.7%) and women (1.0%), together with Spain, the Netherlands, and Romania. Nevertheless, there are more than seven million at-risk consumers, including fully productive adults of both sexes. In particular, at-risk consumers in 2021 comprise 20.0% of men and 8.7% of women over 11 years of age, for a total of over 7,700,000 individuals (M = 5,250,000; F = 2,450,000) [11].

Because of the ubiquitous distribution of this substance in organisms, the harmful effects of exposure to this compound in the mid and long term affect almost all organs and tissues of the body; many of the effects are genetically driven [12,13]. Moreover, findings from human and animal models evidence that pre-conceptional alcohol drinking, both in females and males, causes a deleterious effect on offspring [14,15,16]; prenatal consumption is considered the direct cause of fetal alcohol spectrum disorders (FASDs) and of the more severe fetal alcohol syndrome (FAS) [17,18,19], while male consumption is the source of negative effects on fetal development [19,20,21].

Based on the diagnostic criteria of the DMS-V, alcohol addiction is characterized by the compulsive search for the substance and its consumption, by the losing of control of the amount of substance consumed, and by the arising of a negative emotional state when access to the substance is denied [22]. Beyond the criteria established by the Diagnostic and Statistical Manual of Mental Disorders, in 1985, Lesch and colleagues developed a system to classify alcohol-dependent patients. This system divides alcohol-dependent patients into four categories (from I to IV) according to biochemical, physiological, clinical, and behavioral characteristics. Specific features of alcohol-dependent patients assign each individual to a certain category. According to Lesch’s classification, Type I patients drink alcohol to counteract symptoms of withdrawal. Type II patients use alcohol as a conflict-solving and anxiety-reducing agent. Type III patients ingest alcohol to ‘self-medicate’ to address affective disorders (alcohol is used as an antidepressant). Type IV patients have a history of cerebral impairment that precedes the development of alcohol dependence. The differences in alcohol use history are not negligible, because each different motivation for craving and relapse may be treated differently in terms of pharmacological approach and supportive psychological therapy [23,24]. Moreover, the cluster distribution of alcohol-dependent men and women according to the typology of Lesch in a Bulgarian population of 140 alcohol-dependent individuals of both sexes evidenced the need to modify conventional treatment and therapeutic protocols to provide more suitable responses to the tendency of an increasing number of alcohol-dependent women in this country, managing also the relevant gender-related differences that have to be considered in a psychotherapeutic approach [25]. Furthermore, the comorbidity of AUD with psychiatric illnesses and the entwined relations between these conditions raise questions still unanswered in terms of diagnostics and therapeutic interventions [26].

The various stages of dependence are linked to defects of different neuronal circuits and brain areas and also to the action of releasing factors from the hypothalamus. It should be noted that genetic heritage plays an important role in terms of alcohol consumption and dependence [27,28,29]. Indeed, a large body of evidence sheds light on the role of genetics as a risk factor in AUD [30,31,32,33,34]. Moreover, genetic variants can condition not only the metabolic pathways of ethanol but also the neurobiology of alcohol dependence, which is mediated by the neuronal circuits that regulate reward, pleasure, desire, and impulsiveness [35,36,37,38]. It has been found that 5-HTT, a molecular target of many antidepressant drugs, plays a pivotal role in serotonergic transmission and is associated with many psychiatric diseases as well as alcohol addiction [39,40,41].

Transcriptional regulation of 5-HTT depends not only on epigenetic modifications, among which DNA methylation (CpG and non-CpG) represents the most important mechanism [42,43], but also on sequence variations that can occur in coding and non-coding regions, in intron–exon boundaries crucial for the splicing machinery, and 5′ and 3′ untranslated regions (UTR) that are important, respectively, for the binding of transcription factors and micro RNAs [44]. Moreover, these sequence variations not only have a role in the dynamics of gene expression but also may alter the response to pharmacological therapies that target 5-HTT or serotonin receptors [45,46,47].

Among the large number of sequence variations that act on the expression of the SLC6A4 gene (coding for 5-HTT) or on the function of 5-HTT, the more studied are the serotonin transporter-linked polymorphic region 5-HTTLPR and the single nucleotide polymorphism (SNP) rs25531; both occur in the 5′UTR of the SLC6A4 gene [48,49,50]. The serotonin transporter-linked polymorphic region 5-HTTLPR (Long or Short allele, respectively having 14 or 16 repeats) and the SNP rs25531 (A or G allele) act together as a tri-allelic polymorphism (La-Lg-S alleles; Lg and S with low function) [48,51,52,53].

A full knowledge of the molecular genetics of alcohol dependence is essential for a greater understanding of the mechanisms that produce phenotypic variability in alcohol-addicted individuals.

This evidence induced us to evaluate the allelic and genotypic frequencies of 15 SNPs and two variable number of tandem repeat (VNTR) regions, known to affect the expression and/or the function of 5-HTT (Table 1), comparing a population of alcohol-dependent subjects to a control group of abstemious individuals. Therefore, based on our results, we performed a haplotype and linkage analysis in the populations covered by our study.

## 2. Results

### 2.1. Allelic Frequencies

Table 2 shows the allelic frequencies of each polymorphism and VNTR, in terms of number of individuals with a certain allele and relative frequencies in AUD subjects and controls. Statistical analysis evidenced a difference between alcohol-dependent subjects and healthy controls, in allelic frequencies concerning the tri-allelic polymorphism (*p* = 0.0083, over the limit of the Bonferroni’s correction; see methods), with the higher function alleles more represented in the control population. Low-function alleles (Lg and S) were considered cumulatively. Allelic frequencies concerning the other SNPs studied were not statistically significant.

### 2.2. Genotipic Frequencies

Table 3 shows the genotypic frequencies of each polymorphism and VNTR, in terms of the number of individuals with a certain genotype and relative frequencies, in AUD subjects and controls. A statistically significant difference between alcohol-dependent subjects and healthy controls was found also in genotypic frequencies concerning the tri-allelic polymorphism (*p* = 0.0151, over the limit of the Bonferroni’s correction; see methods), with the higher function genotypes more represented in the control population (Table 3). Genotypes with at least one high-function allele (La-Lg and La-S) and genotypes with low-function alleles (Lg-Lg, Lg-S, and S-S) were considered cumulatively. Genotypic frequencies concerning the other SNPs were not statistically significant.

### 2.3. Haplotype Analysis

Haplotype analysis was performed, selecting common haplotypes with a frequency above 5% in at least one of the two categories of the subjects studied. Table 4 shows the number of times and the relative frequency the selected haplotype was found in AD subjects or controls, versus all the other haplotypes discovered by the analysis in the specific population. We found that only the haplotype H5 was more frequent in the control population (*p* < 0.0001). All the other haplotypes, H2, H3, and H4, were found to be more frequent in AUD individuals (*p* < 0.0001), including the haplotype H1, whose results were not statistically significant after Bonferroni correction.

We found that twelve variations did not differ between H2, H3, H4, and H5 haplotypes. Ten of these variations were present in the four haplotypes as “loss of function” alleles, with two as “gain of function”. Five variations, namely rs25531, rs1042173, rs25532, 5-HTTLPR, and STin2, were found to be different in the four significant haplotypes. In Table 5, we describe how these variations are functionally combined within the considered haplotypes. H2, H3, and H4 (more frequent in AUD) showed a prevalence of gain of function alleles when compared to the H5 haplotype (more frequent in the control population). Moreover, the polymorphism rs25532 was always present as a “gain of function” allele in the haplotypes that were more frequent in AUD individuals, while it occurred as a “loss of function allele” in the H5 haplotype.

## 3. Discussion

Reviewing the literature concerning the genetic predisposition to AUD, it should be noted that a large number of studies analyze a few genetic variations [68] or even a single polymorphism [69] suspected to affect the activity of a possible “disease-linked” gene. This kind of study, which evaluates allelic and genotypic frequencies of one or more sequence variations between a population of affected individuals compared to healthy controls, namely case–control studies (CCSs), saw a great increase starting from the early 1990s. With the exception of Genome-Wide Association Studies (GWASs), which simultaneously analyze thousands of genomic variants on a “hypothesis-free” basis, many CCSs evaluate the effect of a small number of genome variations that are possibly engaged in the etiology of a certain disease.

Frequently, the traits considered derive from other studies where these variations have been evaluated in the context of another pathology and their effect has been ascertained. Sometimes, GWAS studies represent the source of new susceptibility loci to further investigate in independent studies. The presence of a certain variation can exert a “loss of function” effect due to a lower expression of the gene or an impaired function of the coded protein or, for opposite reasons, to a “gain of function”. Moreover, it must be considered that a variant with a “negative” effect can be counterbalanced by a variant with a “positive” effect within the same gene. Many inconsistent reports of associations between functional variations (for example, 5HTTLPR) and susceptibility to depression and anxiety disorders or alcohol dependence have been reported to date. One of the reasons could be that it is not known how many other sequence variants might contribute to the transcriptional variation of the serotonin transporter gene, nor whether their presence might confound the interpretation of 5HTTLPR in genetic association studies. Furthermore, another confounding factor could be the partial or incorrect classification of patients in terms of alcohol dependence and comorbidity with other severe mental illnesses. A more strict and careful stratification of patients (following the Lesch criteria, for example) could be of help in understanding the role of each genetic variation and the molecular mechanisms involved in alcohol addiction.

In our study, through a deep review of the literature and the analysis of the human gene mutation database (HGMD—https://www.hgmd.cf.ac.uk, accessed on 28 May 2024), we selected all the variations of the SLC6A4 gene known to have an effect (positive or negative) on the encoded protein, the serotonin transporter, in terms of expression and/or function, with the evaluated variants in a population of alcohol-dependent individuals and a control group of healthy individuals matched by age and demographic characteristics.

We found a statistically significant difference in allelic and genotypic frequencies concerning rs25531 e 5-HTTLPR, considered a tri-allelic polymorphism, with the “low function” alleles more frequent in the AUD population. We suppose that the presence of the low-function alleles can affect the reuptake of serotonin at the synaptic level, leading to an increased permanence of 5-HTT in the intersynaptic space and to a reinforced effect of the rewarding properties of the alcohol. Interestingly, in a previous study conducted by our group [48], in a smaller group of AUD patients compared to healthy controls, we found no significant differences concerning these two variations. This means that, to ensure the state of the art of this kind of study, the numerosity of the case series is a fundamental requirement for retrieving reliable and translational information. Moreover, the involvement of 5-HTTLPR in alcohol addiction and major depression was confirmed by a recent meta-analysis that included a very large number of cases and controls, confirming the homozygous S allele as an increased risk factor [70].

A particular strength of our study was also the haplotype-based approach that allowed us to obtain information about common haplotypes within the SLC6A4 gene that have shown a strong association of the SLC6A4 gene variability with alcohol dependence. Notably, the only haplotype more frequent in the control population carries the G allele of the rs25531 polymorphism together with the S allele of 5-HTTLPR, as well as two of the four haplotypes more frequent in AUD individuals, while the other two carry the A allele of the rs25531 SNP together with the L allele of 5-HTTLPR VNTR. Possibly, the effect of the 5-HTTLPR tri-allelic polymorphism alone is not enough to discriminate the two populations analyzed in the study. As a matter of fact, the analysis of a single trait, like the 5-HTTLPR/rs25531 tri-allelic polymorphism, shows conflicting results with the analysis of the haplotypes. In the first case, we have a prevalence of “loss of function” alleles (Lg-S) in the AUD population (either as allelic or as genotypic frequencies).

Considering instead the analysis of the haplotypes, we have H2, H3, and H4 (more frequent in AUD) that show an increased number of “gain of function” alleles, although this balance still remains in favor of a greater number of those deemed “loss of function”. This balance between SNPs with different functions possibly modulates the overall functionality of the haplotype. This scenario opens another possible interpretation, where the “gain of function” of the serotonin transporter could indirectly lead to increased alcohol consumption to obtain the desired rewarding effect. This apparent contrast may be due to the multifactorial character of the disease, where the intragenic variability can concur with the polygenic nature of this condition. Interestingly, all the variations that differ in the H2, H3, H4, and H5 haplotypes are in non-coding regions of the SLC6A4 gene: rs25531, 5-HTTLPR, and rs25532 in 5′UTR; STin2 in intron 2; and rs1042173 in 3′UTR. These variations are important for the creation of consensus binding sequences for transcription factors (STin2 and rs25531) and for the interaction with other promoter regulatory regions (5-HTTLPR). The variation rs1042173 is located not only at a putative polyadenylation signal site in 3′UTR of the SLC6A4 gene but also near a potential binding site for microRNA miRNA-135 [56,71,72].

Our results for genotyping and haplotype analysis are consistent with the role of the SLC6A4 gene variability in alcohol dependence, and what we saw for the tri-allelic polymorphism confirms what is already present in a large part of the literature about this topic; the role of the SLC6A4 gene haplotypes should be clarified and needs further studies. One strategy could be that of using constructs that contain the variations constituting the haplotypes evidenced, to study their impact on the expression of the SLC6A4 gene. Our preliminary findings shed light on the role of SLC6A4 sequence variations in alcohol dependence, but further investigations are needed to understand the complexity of genetics in AUD.

In conclusion, based on our data, we speculate that 5-HTT genetic variations significantly impact the susceptibility to alcohol addiction. The detected differences in allelic and genotypic frequencies of the tri-allelic polymorphism, with higher function alleles being more predominant in the controls, emphasize the protecting role of some genetic variants against alcohol abuse. Furthermore, the discovery of specific haplotypes associated with alcohol addiction implies a composite genetic interplay contributing to AUD. These findings highlight the consequence of considering genetic factors in alcohol addiction and indicate potential targets for therapeutic intervention. However, additional research is necessary to elucidate the subtle mechanisms by which these haplotypes influence 5-HTT function and expression in the framework of alcohol addiction.

## 4. Materials and Methods

### 4.1. Analyzed Populations and Sampling

We genotyped a total of 1447 patients and 441 controls. An informed consent was administered to all patients and controls. The patients, 74% men (n = 1071) and 26% women (n = 376), were alcohol-dependent subjects, mainly of Italian origin, being treated at the Alcohol Reference Center of the Lazio region at the Policlinico Umberto I University Hospital in Rome. The patients were between 18 and 65 years old. The T-ACE questionnaire was administered to the patients who were referred to the Center. Moreover, they were all subjected to a careful psychiatric examination. The exclusion criteria for the recruitment of the alcohol-dependent people included history of head injury, loss of consciousness, history of organic mental disorder, present consumption of psychoactive drugs (such as cocaine, opioids, amphetamine, other recreational drugs, anxiolytics, euphoriants, antipsychotics, barbiturates, antidepressants, and hallucinogens; data based on urine toxicology), seizure disorder or central nervous system diseases, and signs of hypertension at the time of recruitment. A detailed description of the alcohol-dependent subjects considered in our study is given in Table 6. The control population of healthy blood donors, from the Policlinico Umberto I Transfusion Center, was matched for age, sex, and ethnic origin (the only info available for controls). In addition to the informed consent, each healthy donor was asked to fill in a personal data sheet with information on lifestyle, with particular regard to habits relating to the abuse of alcoholic beverages (type of drink and frequency of consumption), tobacco consumption, the abuse of drugs (e.g., repetitive consumption of certain drugs), and the possible consumption of narcotic substances. It should be noted that a careful check of the blood analyses was carried out for every donation.

For DNA extraction, 5 mL of peripheral blood was collected from each participant in BD Vacutainer™ tubes with EDTA as an anticoagulant (BD, Franklin Lakes, NJ, USA). The blood was then stored at −80 °C up until the day of processing. Blood samples were drawn from patients at their first visit to the Alcohol Reference Center of the Lazio region at the Policlinico Umberto I University Hospital in Rome.

### 4.2. DNA Extraction

DNA extraction was performed from 2 mL of whole blood from the previously collected EDTA tubes, using the QIAampDNA Mini Kit (Qiagen, Manchester, UK), based on a chromatographic system with single-use columns. After extraction, DNA was quantified by the Qubit fluorimetric assay (Invitrogen, Carlsbad, CA, USA).

### 4.3. SNP and VNTR Region Genotyping

A total of 15 SNPs and 2 VNTR regions were investigated in the AUD population and controls (Figure 1).

Overall, 15 PCR amplicons were generated for the further analysis of the SNPs and of the VNTR regions (Table 7).

### 4.4. Polymerase Chain Reaction Assays

Two multiplex PCRs were designed to amplify the DNA of the regions surrounding the variations of interest (Figure 2).

Polymerase Chain Reactions (PCRs) were performed in a PTC100 Thermal Cycler (Bio-Rad, Hercules, CA, USA) in a reaction volume of 15 µL. The reaction mix was assembled as follows: 3.35 µL of tetradistilled water, 2.1 µL of deoxynucleotides (dNTP) 1.25 mM (Thermo Fisher Scientific, Waltham, MA, USA), 3 µL of Buffer Mix Re-action 5× (Promega, Madison, WI, USA), 0.9 µL of magnesium chloride 25 mM (Promega), 0.6 µL of the primer mix (MIX1 or MIX2), 0.05 µL of Gotaq DNA polymerase 5 U/μL (Promega), and 5 µL of genomic DNA (10 ng). The PCR protocol included the following steps: 95 °C for 2′; (94 °C for 45″; 64 °C for 1′ 30″; 72 °C for 2′ 30″) for 28 cycles; 72 °C for 7′; 10 °C for ∞. Amplicons were distinguished by their size in agarose gel electrophoresis using a 50 bp molecular weight standard (Thermo Fisher), running 3.3 µL of each sample in a 1.5% gel (Bio-Rad) (Figure 2).

An enzymatic purification was performed for each sample, adding to each PCR tube 1.2 μL of FastAP (Thermosensitive Alkaline Phosphatase), 0.6 μL of Exonuclease I, and 1.5 μL of 10× Reaction Buffer for Exonuclease I buffer (all from Thermo Fisher) in a total volume of 15 μL. Samples were then incubated at 37 °C for 60′ (activation of the enzymes) and then at 80 °C for 15′ (enzyme deactivation).

### 4.5. Mini-Sequencing Assay

The SNPs analysis was performed using the mini-sequencing reaction, a single base primer extension technique. Specific mini-sequencing probes flanking the 3′ end of the sequence variations were used to investigate each SNP (Table 8). All the oligos (primers and probes) were designed using as a template the reverse strand of the SLC6A4 human gene sequence (ENSG00000108576.10 (SLC6A4)) obtained from the database Ensebl (https://www.ensembl.org/ accessed 28 May 2024). The reaction mixture was prepared using the SNaPshot Multiplex Ready Reaction Mix (MRRM) (Thermo Fisher). The reaction mix was prepared by adding 2.5 μL of MRRM, 0.5 μL of the SNaP primer mix (MIX1 or MIX2), and 2 μL of the purified PCR product, for a total volume of 5 μL. The solution was then placed in the thermal cycler and underwent the following steps: (96 °C for 10″; 50 °C for 5″; 60 °C for 30″) for 30 cycles; 10 °C for ∞. At the end, the mini-sequencing reaction was again purified to remove unincorporated ddNTPs, adding 0.5 μL of FastAP (Thermo Fisher) and 0.5 μL of distilled water to 1 μL of the mini-sequencing reaction with a two-step incubation: 37 °C for 60″ and 80 °C for 15′. Two microliters of purified sample were then incubated at 95 °C for five minutes with 0.5 μL of GeneScan 120 LIZ Size standard (Thermo Fisher) and 7.5 μL of formamide (Thermo Fisher) as a denaturating agent. Fragment separation was performed by capillary electrophoresis on an ABI PRISM 3130*xl* genetic analyzer with a 36 cm capillary array (Thermo Fisher), using the POP6 polymer. Migration results were analyzed with GeneMapper v.4.1 software (Thermo Fisher).

### 4.6. VNTR Region Genotyping

Patients and controls were genotyped for the two VNTR regions (5-HTTLPR and STin2) through an electrophoretic analysis of the PCR amplicons. 5-HTTLPR 16 or 14 repeats resulted in a 43 bp variation (respectively insertion/deletion) in DNA sequence [73]. The PCR amplification of the 17 bp VNRT polymorphism, namely STin2, in the second intron of the 5-HTT, generated three amplicons of different sizes: 350 bp for the STin2.12 allele, 320 for the STin2.10, and 305 for the STin2.9. The reaction components were the same as described for the amplification of the regions surrounding the SNPs, but some of the PCR conditions were different: 0.6 µL of each primer (5-HTTLPR forward and reverse, or STin2), 2.75 µL of tetra-distilled water, annealing temperature of 64 °C, and 40 cycles. An example of 5-HTTLPR genotyping is reported in Figure 3.

### 4.7. Haplotype Analyses

The FamHap software (V16) was used for haplotype analysis in the general population and patients for SLC6A4 gene polymorphisms [74].

### 4.8. Statistical Analysis

The chi-square test (Cochran-Armitage test: https://www.graphpad.com/support/faq/the-chi-square-test-for-trend/ accessed on 28 May 2024) was applied to evidence eventual significant differences between the two populations studied. Bonferroni correction was applied for all allelic and genotypic frequencies (*p* = 0.05/17 = 0.003) and all haplotype analyses (*p* = 0.05/111 = 0.0005). The statistical analysis was performed using GraphPad Prism 5.01 for Windows (GraphPad Software, Boston, MA, USA).

## Figures and Tables

**Figure 1 ijms-25-08089-f001:**
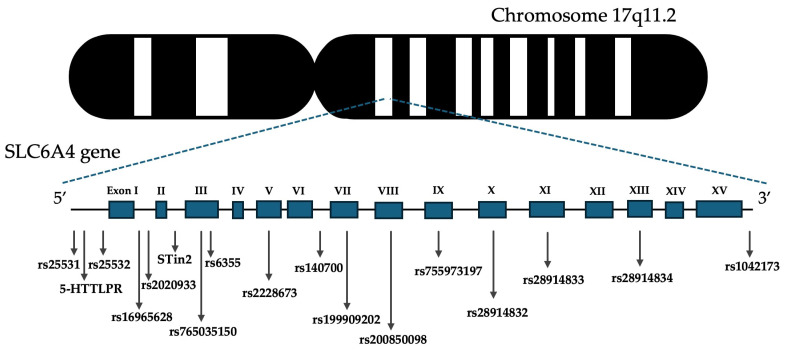
Genomic localization of SLC6A4 polymorphisms examined in this study.

**Figure 2 ijms-25-08089-f002:**
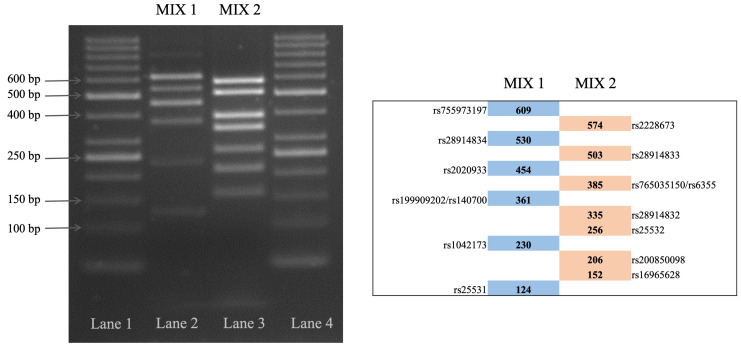
Gel image and schematic view of two multiplex PCRs. Left panel: Lanes 1 and 4, 50 bp Ladder; Lane 2, DNA sample amplified with primer MIX1; Lane 3, DNA sample amplified with primer MIX2. Right panel: schematic representation of amplicons with rs number of SNPs of interest (next to each box) and dimensions in bp (within each box).

**Figure 3 ijms-25-08089-f003:**
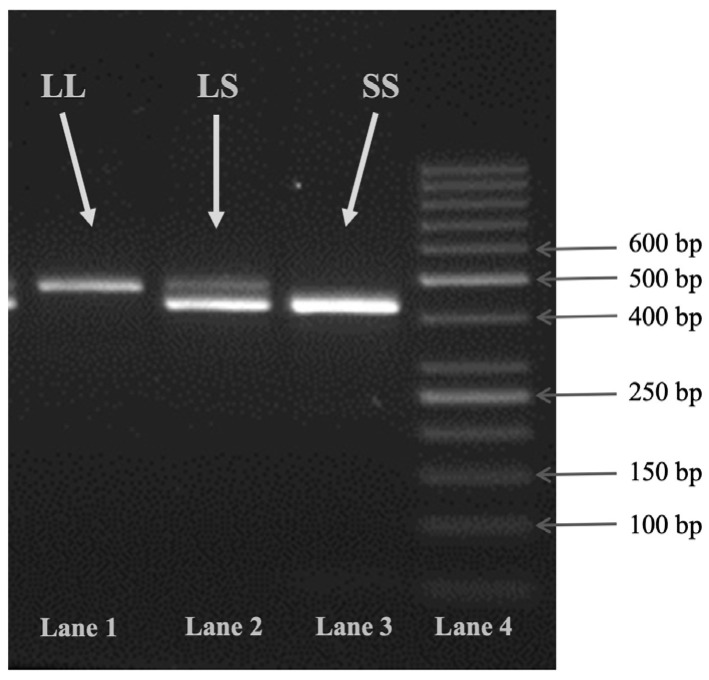
Electrophoresis of 5-HTTLPR. Lane 1: homozygous LL (L allele; 458 bp fragment). Lane 2: heterozygous LS (L and S alleles; 458 bp and 415 bp fragments). Lane 3: homozygous SS (S allele; 415 bp fragment). Lane 4: 50 bp ladder.

**Table 1 ijms-25-08089-t001:** Analyzed SNPs and VNTRs and their functional effect. Variations are in the same order as they are reported in the haplotype.

Variation	Position	Base Shift	Amino Acid Change	Function	References
rs25531	Promoter	SNP A/G		Loss (G allele)	[54]
rs2020933	Intron 1	SNP T/A		Gain (A allele)	[55]
rs1042173	3′-UTR	SNP T/G		Gain (G allele)	[56,57]
rs140700	Intron 6	SNP G/A		Loss (A allele)	[58]
rs199909202	Exon 7	SNP C/T	Ser293Phe	Gain (T allele)	[59]
rs755973197	Exon 9	SNP C/A	Leu362Met	Gain (A allele)	[59]
rs28914834	Exon 13	SNP C/G	Leu550Val	Gain (G allele)	[60,61]
rs25532	Promoter	SNP C/T		Loss (T allele)	[62]
rs16965628	Intron 1	SNP C/G		Gain (G allele)	[62]
rs28914832	Exon 10	SNP A/G	Ile425Val	Gain (G allele)	[59,63]
rs2228673	Exon 5	SNP G/T	Lys201Asn	Gain (T allele)	[64]
rs200850098	Exon 8	SNP C/T	Pro339Leu	Loss (T allele)	[59,65]
rs765035150	Exon 3	SNP A/G	Thr4Ala	Gain (G allele)	[59]
rs6355	Exon 3	SNP G/C	Gly56Ala	Gain (C allele)	[59,61,66]
rs28914833	Exon 11	SNP T/C	Phe465Leu	Gain (C allele)	[60]
5-HTTLPR	Promoter	VNTR 43 bp	14S-16L	Loss (S allele)	[49]
STin2	Intron 2	VNTR 17 bp	9/10/12 repeats	Loss (9 and 10 repeats)	[67]

**Table 2 ijms-25-08089-t002:** Allelic frequencies analyzed in this study: 5-HTTLPR and rs25531 were considered as a tri-allelic polymorphism. AUD, individuals with alcohol use disorders; CI, confidence interval; Ctrls, healthy controls; na, not measurable. The asterisks indicate the statistical significance after chi-square testing (** *p* < 0.01).

Polymorphisms	n Ctrls	n AUD	Allele	Ctrls n alleles (freq.)	AUD n alleles (freq.)	*p* Value	CI 95%
rs1799971	440	1447	A	747 (0.849)	2486 (0.859)	0.4853	0.8775 to 1.341
			G	133 (0.151)	408 (0.141)		
rs2020933	438	1008	T	807 (0.921)	1851 (0.918)	0.8378	0.7156 to 1.286
			A	69 (0.079)	165 (0.082)		
rs1042173	440	1391	T	439 (0.499)	1481 (0.532)	0.0901	0.7514 to 1.018
			G	441 (0.501)	1301 (0.468)		
rs140700	441	1008	C	812 (0.921)	1861 (0.923)	0.8775	0.7713 to 1.389
			T	70 (0.079)	155 (0.077)		
rs199909202	441	1008	C	882 (1.0)	2016 (1.0)	na	na
			T	0 (0.0)	0 (0.0)		
rs755973197	441	968	C	882 (1.0)	1936 (1.0)	na	na
			A	0 (0.0)	0 (0.0)		
rs28914834	441	968	C	882 (1.0)	1936 (1.0)	na	na
			G	0 (0.0)	0 (0.0)		
rs25532	168	215	C	309 (0.92)	388 (0.902)	0.4467	0.4866 to 1.339
			T	27 (0.08)	42 (0.098)		
rs16965628	440	982	C	805 (0.915)	1768 (0.90)	0.2484	0.6358 to 1.111
			G	75 (0.085)	196 (0.10)		
rs28914832	440	983	T	879 (0.999)	1965 (0.999)	0.8562	0.1396 to 35.81
			C	1 (0.001)	1 (0.001)		
rs2228673	441	984	C	882 (1.0)	1968 (1.0)	na	na
			A	0 (0.0)	0 (0.0)		
rs200850098	440	984	C	880 (1.0)	1968 (1.0)	na	na
			T	0 (0.0)	0 (0.0)		
rs765035150	441	944	A	882 (1.0)	1888 (1.0)	na	na
			G	0 (0.0)	0 (0.0)		
rs6355	441	944	C	874 (0.982)	1875 (0.986)	0.7021	0.5451 to 3.198
			G	8 (0.018)	13 (0.014)		
rs28914833	441	944	A	882 (1.0)	1888 (1.0)	na	na
			G	0 (0.0)	0 (0.0)		
rs25531+5-HTTLPR	434	1049	La	447 (0.515)	967 (0.461)	0.0083 **	0.6873 to 0.9435
			Lg-S	421 (0.485)	1131 (0.539)		
STin2	439	1028	9	5 (0.006)	19 (0.009)	0.5273	na
			10	293 (0.334)	706 (0.343)		
			12	580 (0.660)	1331 (0.648)		

**Table 3 ijms-25-08089-t003:** Genotypic frequencies analysis: 5-HTTLPR and rs25531 were considered as a tri-allelic polymorphism. AUD, individuals with alcohol use disorders; CI, confidence interval; Ctrls, healthy controls; na, not measurable. The asterisks indicate the statistical significance after chi-square testing (* *p* < 0.05).

Polymorphisms	n Ctrls	n AUD	Genotype	Ctrls n (freq.)	AUD n (freq.)	*p* Value	CI 95%
rs1799971	440	1447	AA	319 (0.725)	1069 (0.739)	0.6676	na
			GA	109 (0.248)	348 (0.240)		
			GG	12 (0.0270)	30 (0.021)		
rs2020933	438	1008	TT	370 (0.845)	853 (0.846)	0.2843	na
			AT	67 (0.153)	145 (0.144)		
			AA	1 (0.020)	10 (0.010)		
rs1042173	440	1391	TT	117 (0.266)	402 (0.289)	0.1596	na
			TG	205 (0.466)	677 (0.487)		
			GG	118 (0.268)	312 (0.224)		
rs140700	441	1008	CC	374 (0.848)	858 (0.851)	0.9068	na
			CT	64 (0.145)	145 (0.144)		
			TT	3 (0.007)	5 (0.005)		
rs199909202	441	1008	CC	441 (1.0)	1008 (1.0)	na	na
			CT	0 (0.0)	0 (0.0)		
			TT	0 (0.0)	0 (0.0)		
rs755973197	441	968	CC	441 (1.0)	968 (1.0)	na	na
			CA	0 (0.0)	0 (0.0)		
			AA	0 (0.0)	0 (0.0)		
rs28914834	441	968	CC	441(1.0)	968 (1.0)	na	na
			CG	0 (0.0)	0 (0.0)		
			GG	0 (0.0)	0 (0.0)		
rs25532	168	215	CC	142 (0.845)	174 (0.809)	0.6223	na
			CT	25 (0.149)	40 (0.186)		
			TT	1 (0.006)	1 (0.005)		
rs16965628	440	982	CC	366 (0.832)	798 (0.813)	0.1673	na
			CG	73 (0.166)	172 (0.175)		
			GG	1 (0.002)	12 (0.012)		
rs28914832	440	983	TT	439 (0.998)	982 (0.999)	na	na
			TC	1 (0.002)	1 (0.001)		
			CC	0 (0.0)	0 (0.0)		
rs2228673	441	984	CC	441 (1.0)	984 (1.0)	na	na
			CA	0 (0.0)	0 (0.0)		
			AA	0 (0.0)	0 (0.0)		
rs200850098	440	984	CC	440 (1.0)	984 (1.0)	na	na
			CT	0 (0.0)	0 (0.0)		
			TT	0 (0.0)	0 (0.0)		
rs765035150	441	944	AA	944 (1.0)	441 (1.0)	na	na
			AG	0 (0.0)	0 (0.0)		
			GG	0 (0.0)	0 (0.0)		
rs6355	441	944	CC	433 (0.982)	931 (0.986)	na	na
			CG	8 (0.018)	13 (0.014)		
			GG	0 (0.0)	0 (0.0)		
rs28914833	441	944	AA	441 (1.0)	944 (1.0)	na	na
			AG	0 (0.0)	0 (0.0)		
			GG	0 (0.0)	0 (0.0)		
rs25531+5-HTTLPR	434	1049	La/La	116 (0.267)	245 (0.234)	0.0151 *	na
			La/Lg–La/S	215 (0.496)	477 (0.454)		
			Lg/Lg–Lg/S–S/S	103 (0.237)	327 (0.312)		
STin2	439	1028	9/9	0 (0.0)	1 (0.001)	0.8319	na
			10/10	53 (0.121)	120 (0.117)		
			12/12	195 (0.444)	427 (0.415)		
			9/10	1 (0.002)	3 (0.003)		
			9/12	4 (0.009)	14 (0.014)		
			10/12	186 (0.424)	463 (0.450)		

**Table 4 ijms-25-08089-t004:** Haplotype analysis. The variations are in the same order as they are reported in the haplotype. AUD, individuals with alcohol use disorders; df, degree of freedom; CI, confidence interval; OR, odds ratio; The asterisks indicate the statistical significance after chi-square testing (* *p* < 0.05, ** *p* < 0.01, *** *p* < 0.001); in bold are indicated the *p* values still significant after Bonferroni’s correction (*p* < 0.0005).

Specific Haplotype * [*vs* All the Other Haplotypes]	Ctrls n (Frequency)	AUD n (Frequency)	Χ, df	*p*	OR	CI
H1: G T T C C C C C C T C C A C A 14 12	30 (0.033)	106 (0.056)	6.662, 1	0.0098 **	0.5836	0.3860 to 0.8825
all the other haplotypes	868 (0.967)	1790 (0.944)				
H2: A T G C C C C C C T C C A C A 16 12	57 (0.063)	261 (0.138)	33.25, 1	**<0.0001 *****	0.4246	0.3150 to 0.5722
all the other haplotypes	841 (0.937)	1635 (0.862)				
H3: A T T C C C C C C T C C A C A 16 10	81 (0.090)	384 (0.203)	55.43, 1	**<0.0001 *****	0.3904	0.3027 to 0.5034
all the other haplotypes	817 (0.910)	1512 (0.797)				
H4: G T G C C C C C C T C C A C A 14 12	129 (0.144)	405 (0.214)	19.29, 1	**<0.0001 *****	0.6176	0.4974 to 0.7668
all the other haplotypes	769 (0.856)	1491 (0.786)				
H5: G T G C C C C T C T C C A C A 14 12	129 (0.144)	86 (0.045)	82.89, 1	**<0.0001 *****	3.531	2.653 to 4.698
all the other haplotypes	769 (0.856)	1810 (0.955)				

**Table 5 ijms-25-08089-t005:** Functional description of the variations in common haplotypes. H2, H3, and H4 haplotypes are more frequent in AUD individuals, while H5 is more frequent in the control population. The + symbol indicates the allele with “gain of function”; the - symbol indicates the “loss of function” allele.

Variation	Base Shift	Function	H2	H3	H4	H5
rs25531	SNP A/G	Loss (G allele)	+	+	-	-
rs1042173	SNP T/G	Gain (G allele)	+	-	+	+
rs25532	SNP C/T	Loss (T allele)	+	+	+	-
5-HTTLPR	VNTR 43 bp	Loss (S allele)	+	+	-	-
STin2	VNTR 17 bp	Loss (9 and 10 repeats)	+	-	+	+

**Table 6 ijms-25-08089-t006:** Alcohol-dependent patient characteristics. Data are expressed as percentage or SEM.

Study Sample (n = 1447)	
Age in years	45.37 ± 10.05
Ethnic Origin (%)	
Caucasian	93.5
African	2.7
Hispanics	2.6
Asian	1.2
Marital Status (%)	
Single	37.1
Married	33.7
Separated/Divorced	27.4
Widowed	1.8
Qualifications (%)	
Primary School	1.2
Middle School	41.7
High School	45.3
University Degree	11.8
Employment Status (%)	
Workers	60.3
Unemployed	30.5
Retired	9.2
Smokers (%)	75
Daily cigarettes’ numbers	18.5 ± 2.5
Family History of Alcoholism (%)	83.9
From both parents	12.4
From father	30.6
Alcohol Related Variables	
Age of onset of at-risk drinking	24.8 ± 2.7
Years of at-risk drinking	14.4 ± 2.8
Alcohol units’ intake per die 30 days before Day Hospital admission	15.9 ± 2.3
Alcohol preference	
Wine (%)	49.8
Beer (%)	35.6
Spirit (%)	14.6

**Table 7 ijms-25-08089-t007:** Primer pairs for the analyzed SNPs and VNTR regions. Variations are in the same order as they are reported in the haplotype. rs140700 and rs19909202 share the same primer pair; rs765035150 and rs6355 share the same primer pair.

Variation	Forward Primer	Reverse Primer
rs25531	CAACCTCCCAGCAACTCCCTGTA	ATGCTGGGGGGGCTGCAG
rs2020933	TTTTCTTCTGAACTGGGGCTTTTGC	CATCCATATTGGAACGGTCACTGC
rs1042173	GCGTAGGAGAGAACAGGGATGC	TGGGCCCAAAATATTGGACTAGAG
rs140700	TAGTGGGCTCAGAGGTAGTTCTCCTG	CTGCCAATTGGGTTTCAAGTAGAAG
rs199909202	TAGTGGGCTCAGAGGTAGTTCTCCTG	TCTGCCAATTGGGTTTCAAGTAGAAG
rs755973197	TGTGTGGTGGTCATGGCAGTC	TCCCAGGCTCAAGCAATCTTCC
rs28914834	AGTCCCCCAGCCCCACTTTC	AGGTGCCCATCACCACACC
rs25532	CTGCACCCCTCGCAGTATCC	GGCTGAGCGTCTAGAGGGACTG
rs16965628	CCCCAAGCACTGATTGAGAGCAG	ATCACCACCATACATCCGCAACC
rs28914832	AGATGGAAGCCCCACCCTTCC	CCTCACCGTGCTGTCCAAGC
rs2228673	AACGGCAGGGCCACTTTTCC	GGCCGTGGAGCACTTGAGGTAG
rs200850098	CCCCTGCTGTGTTCCAGGTG	CCGTCGGTCCAATCACCTTCC
rs765035150	GAGTCAATCCCGACGTGTCAATCC	ATCCACCTTCTTGCCCCAGGTC
rs6355	GAGTCAATCCCGACGTGTCAATCC	ATCCACCTTCTTGCCCCAGGTC
rs28914833	GAAGTTCTGTCCACGTGTGCTATTTTG	GGAGTAACAACCTCCCCTCCTTTG
5-HTTLPR	CAACCTCCCAGCAACTCCCTGTA	GAGGGACTGAGCTGGACAACCAC
STin2	GGGAGACCTGGGGCAAGAAG	TCAAGAGGACCTACAGCCCATCC

**Table 8 ijms-25-08089-t008:** The 3′ end mini-sequencing probes. In bold, the non-hybridizing tail of polynucleotides was added as an electrophoretic mobility modifier. Variations are in the same order as they are reported in the haplotype. * The probes are designed on the opposite strand.

SNP	Sequence	Primer Length
**rs25531**	**AAAA**TCCCCCCTGCACCCCC	20 (16 + 4)
**rs2020933**	**AAAGAA**ATCAGTTTTGTCCAGAAAAGTGAACC	32 (26 + 6)
**rs1042173**	**GAAAAGAAAAAAAG**GCCATATATTTTCTGAGTAGCATATA	40 (26 + 14)
**rs140700**	**AGAAAAGAAAAAAAAAGGAAAAA**GAAGACCTTGAGAAAGGAGGG*	44 (21 + 23)
**rs199909202**	**AAAGAGAAAAAAAAAAAAAGA**AGCCACCTTCCCTTATATCATCCTTT	47 (26 + 21)
**rs755973197**	**AGAGAAAAAAGAAAAAAGGAAAAAAAGAAAAAAAAAAAAA**GTTTTCCCCTCCAGAGATGCC	61 (21 + 40)
**rs28914834**	**GGAAAAAAAAAAAGAAAAAAAAAAAAAAAAAGAAGAAAAAAAAAAA**GCCATCAGCCCTCTGTTTCTC	67 (21 + 46)
**rs25532**	**AA**CCCATGCACCCCCGG	17
**rs16965628**	GCTAGGGTATGAAGTAGAAAGGCA	24
**rs28914832**	**AAGGAAAAGA**CGTGATTAACATCAGAAAGAAGATGA *	36 (26 + 10)
**rs2228673**	**GAGAAAAAAAAAGAGGAAGGAAAAAAA**AGTTGCCAGTGTTCCAGGAGTT *	49 (22 + 27)
**rs200850098**	**AAAAAAAGAAAAAAAGAGAGAAAGAAAA**CTCAGATCTTCTTCTCTCTTGGTC	52 (24 + 28)
**rs765035150**	**AAAGAAAAAAAAGAGAAAGAGAAGAAAAAGAA**TACTAACCAGCAGGATGGAGACG	55 (23 + 32)
**rs6355**	**GAAGGAAAAGAAAGAAGAAAAAAAAAAAAAAAAAAA**GATAGAGTGCCGTGTGTCATCT *	58 (22 + 36)
**rs28914833**	**GAGAAAGAGGGGAAAGAAAAAGAAGAAAGAAAGGGAGGAAGAAAA**ATGACCACGGCGAGCACGA *	64 (19 + 45)

## Data Availability

Data are available on reasonable request due to ethical reasons.

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
