# Peer review of "DNA Sequence Variations Affecting Serotonin Transporter Transcriptional Regulation and Activity: Do They Impact Alcohol Addiction?"

_ijms, 2024, doi:10.3390/ijms25158089_

Round 1

Reviewer 1 Report

Comments and Suggestions for Authors

This paper presents data on the genetic underpinnings of alcohol addictions. Some points for further revision can be found below:

The first paragraph is not focused on the topic. Please delete.

Authors need to provide a more extended review of the literature. For example, in lines 53-56, authors can mention the differences between sexes and the psychological different profiles for different types of alcohol addiction (for a relevant discussion on this, please discuss: https://www.internationaljournalofcaringsciences.org/docs/51_ivanova_original_10_3.pdf). Another point missing is the cognition of the individuals with AUDs and the connections with other severe mental illness ( doi: 10.1177/1039856217715999. ) These points can be also raised in the discussion as future points for further examination in research regarding the genetics.

Methodology as a separate section is missing. Please add and explain to the reader what was done so if someone wishes to replicate the findings, to know what steps to follow.

Tables must be explained in the main text as they are difficult to follow.

Comments on the Quality of English Language

Moderate English language editing.

Author Response

Answers to the criticisms raised by reviewer 1

This paper presents data on the genetic underpinnings of alcohol addictions. Some points for further revision can be found below:

Reply: We do thank the reviewer for allowing us to improve the quality of the paper thanks to her/his useful comments.

  1. The first paragraph is not focused on the topic. Please delete.

Reply: As suggested, we deleted the first paragraph and extensively modified the introduction (revised text in red).

  1. Authors need to provide a more extended review of the literature. For example, in lines 53-56, authors can mention the differences between sexes and the psychological different profiles for different types of alcohol addiction (for a relevant discussion on this, please discuss: https://www.internationaljournalofcaringsciences.org/docs/51_ivanova_original_10_3.pdf). Another point missing is the cognition of the individuals with AUDs and the connections with other severe mental illness (doi: 10.1177/1039856217715999.) These points can be also raised in the discussion as future points for further examination in research regarding the genetics.

Reply: As requested, we extended the review of the literature and mentioned the differences between sexes and the psychologically different profiles for different types of alcohol addiction, considering the suggested publications (in red, lines 56-76). Moreover, we raised in the discussion the points suggested (in red, lines 213-217).

  1. Methodology as a separate section is missing. Please add and explain to the reader what was done so if someone wishes to replicate the findings, to know what steps to follow.

Reply: As suggested, we modified the materials and methods section, providing a more detailed description of the steps performed, to facilitate the replication of the experiments (please see the modified text in red).

  1. Tables must be explained in the main text as they are difficult to follow.

Reply: According to the comment of the reviewer, we added a description for each table in the main text (text highlighted in red).

  1. Moderate English language editing.

Reply: As suggested, a further revision of the English language was carried out.

Reviewer 2 Report

Comments and Suggestions for Authors

The article “DNA Sequence Variations Affecting Serotonin Transporter Transcriptional Regulation and Activity: Do They Impact Alcohol Addiction?” by Giampiero Ferraguti and colleagues raises the important issue of the role of genetic predispositions in the development of alcohol addiction. After reading the manuscript, several doubts and comments arise:

1. The introduction is generally well written, but there is no data on the prevalence and morbidity of ADU in the Italian population, which is so important in chronic diseases.

2. In line 88 it is written that the control group consists of people abstaining from alcohol. I am asking the authors to explain this issue what criteria they have adopted for abstinence from alcohol. Was there any key or examination of this group by a psychiatrist.

3. Why did the authors not apply the Bonferroni correction in Tables 2, 3 and 4 despite statistically testing the same group of people many times.

4. Whether the Cochran-Armitage test for trend was used for the analysis of Genotype Frequencies.

5. The description of the statistical methods does not specify what statistical tests the authors used.

6. The section method and material requires supplementation. Although the authors described the genetic sequencing method well, the paragraph regarding the description of obtaining material from addicted people and the control group raises many doubts, e.g.:

- What was the distribution of characteristics such as age, demographic characteristics, education, working/not working in the study participants and the control group?

-How was the material obtained, or only during, for example, detoxification treatment?

- What were the exclusion criteria from the study?

-Was there any clinical test used to measure addiction?

-Is there information about alcohol addiction in the family and whether there were any traumatic experiences in your life history?

-What was the proportion of men and women in the study group and the control group?

-Were the study group and the control group ethnically similar?

-What was the drinking pattern, etc.

-Were there other mental illnesses or co-dependencies in the study participants apart from alcohol addiction?

7. No conclusion.

8. The study results and conclusions are not described in the abstract.

Author Response

Answers to the criticisms raised by reviewer 2

The article “DNA Sequence Variations Affecting Serotonin Transporter Transcriptional Regulation and Activity: Do They Impact Alcohol Addiction?” by Giampiero Ferraguti and colleagues raises the important issue of the role of genetic predispositions in the development of alcohol addiction. After reading the manuscript, several doubts and comments arise:

Reply: We do thank the reviewer for the useful comments aiming to improve our manuscript.

  1. The introduction is generally well written, but there is no data on the prevalence and morbidity of ADU in the Italian population, which is so important in chronic diseases.

Reply: As suggested, the introduction was modified and data about the prevalence of AUD in Italy were added (text highlighted in red, lines 39-44).

  1. In line 88 it is written that the control group consists of people abstaining from alcohol. I am asking the authors to explain this issue what criteria they have adopted for abstinence from alcohol. Was there any key or examination of this group by a psychiatrist

Reply: Following the comment of the reviewer, in the materials and methods section we explained that healthy blood donors were investigated for their habits of alcohol use by a questionnaire administered at the moment of joining the study (in red, lines 286-311). Moreover, blood donors are strongly motivated, and they perceive their donation, which is unpaid, as a real mission. Therefore, they are very careful about different aspects of their health.

  1. Why did the authors not apply the Bonferroni correction in Tables 2, 3 and 4 despite statistically testing the same group of people many times.

Reply: we are sorry for the misunderstanding. We performed Bonferroni’s correction for all allelic and genotypic frequencies (P=0.05/17=0.003) and for all haplotype analyses (P=0.05/111=0.0005) (see lines 439-443).

  1. Whether the Cochran-Armitage test for trend was used for the analysis of Genotype Frequencies.

Reply: Yes, the Cochran-Armitage test for trend was used for the analysis of Genotype Frequencies (check lines 439-443).

  1. The description of the statistical methods does not specify what statistical tests the authors used.

Reply: As suggested, paragraph 4.8 concerning the statistical methods has been improved.

  1. The section method and material requires supplementation. Although the authors described the genetic sequencing method well, the paragraph regarding the description of obtaining material from addicted people and the control group raises many doubts, e.g.:

6a. What was the distribution of characteristics such as age, demographic characteristics, education, working/not working in the study participants and the control group?

Reply: as suggested, a further Table 6 has been added to the Methods section dealing with the demographic characteristics of the alcohol-dependent people (no data are available for the control individuals).

6b. How was the material obtained, or only during, for example, detoxification treatment?

Reply: Blood samples were obtained at the first access to the Alcohol Reference Center of the Lazio region at the Policlinico Umberto I University Hospital in Rome (added in the main text, lines 309-311).

6c. What were the exclusion criteria from the study?

Reply: as requested we included in the Methods the exclusion criteria (lines 292-298).

6d. Was there any clinical test used to measure addiction?

Reply: The t-ACE questionnaire was administered to all of the patients who were referred to the Center. Moreover, they were all subjected to psychiatric examination (lines 290-292).

6e. Is there information about alcohol addiction in the family and whether there were any traumatic experiences in your life history?

Reply: Unfortunately, the last info was not available for all the recruited alcohol-dependent individuals because they did not provide those data to the professionals of the Center. Information about alcohol addiction in the family is shown in Table 6.

6f. What was the proportion of men and women in the study group and the control group?

Reply: Men were 74% (n=1071) and women 26% (n=376), the same proportion was adopted for controls. This information was added in the main text (in red, lines 286-288). Controls were matched for this characteristic.

6g. Were the study group and the control group ethnically similar?

Reply: The control population of healthy blood donors, from the Policlinico Umberto I Transfusion Center, was matched for age, sex and ethnic origin (the only info available for controls) (lines 299-301).

6h. What was the drinking pattern, etc.?

Reply: please see Table 6.

6i. Were there other mental illnesses or co-dependencies in the study participants apart from alcohol addiction?

Reply: according to the exclusion criteria individuals with severe mental illnesses or co-dependencies were not considered.

  1. No conclusion.

Reply: As suggested, further conclusions were added at the end of the discussion (lines 274-283).

  1. The study results and conclusions are not described in the abstract.

Reply: as suggested, the abstract has been updated and improved according to the comments of the reviewer (lines 25-31).

Round 2

Reviewer 2 Report

Comments and Suggestions for Authors

Thank you, taking into account my comments. Now the manuscript is written better. I don't make any comments.